# Partial Agonist Activity of Neonicotinoids on Rat Nicotinic Receptors: Consequences over Epinephrine Secretion and In Vivo Blood Pressure

**DOI:** 10.3390/ijms22105106

**Published:** 2021-05-12

**Authors:** Joohee Park, Antoine Taly, Jennifer Bourreau, Frédéric De Nardi, Claire Legendre, Daniel Henrion, Nathalie C. Guérineau, Christian Legros, César Mattei, Hélène Tricoire-Leignel

**Affiliations:** 1University of Angers, INSERM U1083, CNRS UMR 6015, MITOVASC, SFR ICAT, 49000 Angers, France; joohee.park@univ-angers.fr (J.P.); jennifer.bourreau@inserm.fr (J.B.); frederic.denardi@gmx.fr (F.D.N.); claire.legendre@univ-angers.fr (C.L.); daniel.henrion@univ-angers.fr (D.H.); Nathalie.Guerineau@igf.cnrs.fr (N.C.G.); christian.legros@univ-angers.fr (C.L.); 2Theoretical Biochemistry Laboratory, Institute of Physico-Chemical Biology, CNRS UPR 9080, University of Paris Diderot Sorbonne Paris Cité, 75005 Paris, France; antoine.taly@ibpc.fr; 3IGF, University of Montpellier, CNRS, INSERM, 34000 Montpellier, France

**Keywords:** neonicotinoids, acetamiprid, clothianidin, *α*3*β*4 nAChR, epinephrine secretion, blood pressure, nicotine, acute intoxication

## Abstract

Neonicotinoid insecticides are nicotine-derived molecules which exert acute neurotoxic effects over the insect central nervous system by activating nicotinic acetylcholine receptors (nAChRs). However, these receptors are also present in the mammalian central and peripheral nervous system, where the effects of neonicotinoids are faintly known. In mammals, cholinergic synapses are crucial for the control of vascular tone, blood pressure and skeletal muscle contraction. We therefore hypothesized that neonicotinoids could affect cholinergic networks in mammals and sought to highlight functional consequences of acute intoxication in rats with sub-lethal concentrations of the highly used acetamiprid (ACE) and clothianidin (CLO). In this view, we characterized their electrophysiological effects on rat α3β4 nAChRs, knowing that it is predominantly expressed in ganglia of the vegetative nervous system and the adrenal medulla, which initiates catecholamine secretion. Both molecules exhibited a weak agonist effect on α3β4 receptors. Accordingly, their influence on epinephrine secretion from rat adrenal glands was also weak at 100 μM, but it was stronger at 500 μM. Challenging ACE or CLO together with nicotine (NIC) ended up with paradoxical effects on secretion. In addition, we measured the rat arterial blood pressure (ABP) in vivo by arterial catheterization. As expected, NIC induced a significant increase in ABP. ACE and CLO did not affect the ABP in the same conditions. However, simultaneous exposure of rats to both NIC and ACE/CLO promoted an increase of ABP and induced a biphasic response. Modeling the interaction of ACE or CLO on α3β4 nAChR is consistent with a binding site located in the agonist pocket of the receptor. We present a transversal experimental approach of mammal intoxication with neonicotinoids at different scales, including in vitro, ex vivo, in vivo and in silico. It paves the way of the acute and chronic toxicity for this class of insecticides on mammalian organisms.

## 1. Introduction

Since their introduction on the agrochemical market three decades ago, neonicotinoid insecticides are applied on a wide range of crops in 120 countries thanks to: (i) their efficacy against many pest insects; and (ii) their predicted lower mammalian toxicity [1,2,3,4]. Their application in seed-coating due to their systemic properties shifted production and sales towards large scale and led to them being an obvious choice for field crops such as maize, cotton and soybean throughout the world [2,5]. However, several limitations to their use have emerged over the past fifteen years regarding environmental issues [6,7]. In addition to being highly toxic to non-target insects such as pollinators, they are persistent in soil and water, where their half-life reaches up to 1000 days [8,9,10]. They have been claimed to be insect-selective, but recent studies have shown that vertebrates are also affected by sublethal concentrations of these compounds (see reviews [11,12,13]). Thus, some neonicotinoids cause alterations in reproductive function and abnormal embryonic development in birds [14,15,16,17]. Aquatic animals such as amphibians and fish also appear to be exposed and affected by these molecules [18,19,20]. Neonicotinoids have also been shown to cause a wide range of neurobehavioral effects in mammals [21,22,23] and exert endocrine disruption in deer exposed to field-relevant doses [24]. Thus, there are reasons for concern about neonicotinoid toxicity in vertebrate animals and humans, due to a broad mode of action: they target nicotinic acetylcholine receptors (nAChRs), present in both vertebrates and invertebrates [25]. The data of high concern are the detection of these insecticides and their metabolites in regularly eaten food such as apples, oranges, potatoes, cucumbers and honey [26,27,28]. Hence, their persistence in the environment and their presence in food significantly increase the probability of human exposures, as confirmed by urine detection [29,30] with deleterious effects on the human health [6,28]. Among health issues, acute intoxications with neonicotinoids are correlated with alteration of cardiovascular parameters such as arterial blood pressure (ABP) and heart rate [31,32].

Neonicotinoid synthesis uses nicotine (NIC)–or nithiazine–as a molecular template (Figure 1). NIC is an alkaloid extracted from tobacco and mainly contributes to smoking-induced cardiovascular diseases [33]. Indeed, NIC releases epinephrine from the adrenal medullary tissue by activating nAChRs of the sympathetic nervous system, which acutely increases myocardial contraction and vasoconstriction. Then, heart rate and ABP increase as much as 10–15 bpm and 5–10 mmHg in human, respectively [34]. The main hypertensive effects of NIC have been shown to be mediated through the activation of α3β4 nAChRs. The α3β4 antagonist hexamethonium (HEX) selectively inhibits the NIC-induced ABP increase [35]. We therefore considered that the main effect of NIC on ABP occurs through the α3β4 nicotinic receptor which might be the main target of neonicotinoids. To address this hypothesis, we focused on two neonicotinoid compounds (Figure 1): clothianidin (CLO), no longer approved in the EU for seed treatment but very persistent in soil (half degradation time DT${}_{50}$ range 143–1001 days), and acetamiprid (ACE), which is still approved for orchards and non-persistent in soil (DT${}_{50}$ range 0.8–5.4 days) [36,37,38,39]. We evaluated the acute effects of both insecticides on rat nAChR currents, epinephrine secretion from adrenal medulla and rat systemic blood pressure. In addition, we built a docking model of ACE and CLO on α3β4 nAChRs. Our study highlights the acute effects of neonicotinoids on mammal physiology and underlines the deleterious consequences they might exert chronically.

## 2. Results

### 2.1. Effect of Neonicotinoids on ACh-Evoked Cholinergic Currents

Xenopus oocytes were injected with α3 and β4 subunit RNAs (1:1) to express the functional corresponding receptors. They were challenged with increasing concentrations of each ligand (Figure 2A). CLO and ACE could not be used at concentrations upper than 20 μM. Shown are the dose–response curves. As the natural neurotransmitter, ACh activated α3β4 receptors in a concentration-dependent manner, with an EC${}_{50}$ = 8.43 ± 0.22 μM (Figure 2B). Hexamethonium (HEX), a non-competitive α3β4 nAChR antagonist [40], inhibited ACh-induced currents (Figure 2B, inset). As a classical agonist of nAChRs, NIC activated α3β4 receptors with an EC${}_{50}$ = 4.62 ± 0.22 μM. As expected, ACh and NIC exerted a comparable agonist effect on rat α3β4 receptors expressed in Xenopus oocytes. We then challenged α3β4 subunits-expressing oocytes to increasing concentrations of ACE or CLO. In our hands, both compounds acted as partial agonists, since the currents recorded were significantly smaller than ACh- or NIC-evoked currents (CLO: EC${}_{50}$ = 8.39 ± 0.77 mM; ACE: EC${}_{50}$ = 0.130 ± 0.017 M). Together, our data show that neonicotinoids are weak agonists of rat α3β4 nAChRs expressed in Xenopus oocytes (Figure 2C). In addition, the electrophysiological effects of neonicotinoids on mammalian nAChRs are shown in Appendix A. They act as weak agonists of human α4β2 receptors, and strong agonists of rat α7 receptors [41,42]. Different nAChR subtypes are then possibly targeted by neonicotinoids. We restricted our study to the α3β4 receptor because of its crucial role in the peripheral nervous system.

### 2.2. Effects of Neonicotinoids on Epinephrine Secretion in Rat

This result prompted us to investigate the effect of CLO and ACE on epinephrine secretion from rat adrenal glands. It is established that the secretion of epinephrine is chiefly driven by the release of ACh at the splanchnic nerve-chromaffin cell synapses, and the role of α3β4 nAChRs has been evidenced for a long time in this process [43]. We hypothesized that the partial agonist effect of CLO and ACE on these receptors might either: (i) evoke a secretory response in the adrenal gland; or (ii) disrupt the NIC-induced secretion. We took advantage of an ex vivo technique that we designed previously [44] to measure epinephrine secretion (Figure 3A). As expected, NIC dose-dependently stimulates epinephrine secretion, and this effect is inhibited by HEX, confirming the involvement of α3-containing nAChRs such as α3β4, in this secretory activity (Figure 3B). Similar results were obtained for NIC-, CLO- or ACE–induced norepinephrine secretion (data not shown). The minimum NIC concentration eliciting an epinephrine secretion was 3 μM (*p* < 0.05) and the maximal effect was observed in this assay with 100 μM NIC (*p* < 0.05). At the same concentration of 100 μM, neither ACE nor CLO stimulated an epinephrine secretion. However, at 500 μM, CLO or ACE elicited a significant secretory effect (ACE or CLO vs. CTRL *p* < 0.01), which was inhibited by HEX (ACE 500 μM ± HEX *p* < 0.005; CLO 500 μM ± HEX *p* < 0.05) (Figure 3C,E). ACE-induced secretion was not significantly different at 100 and 500 μM, whereas CLO-induced secretion at 500 μM was significantly higher than 100 μM, reflecting their agonist properties shown in Figure 2. Interestingly, this potentiation in secretion was abolished when NIC (10 μM) was added (Figure 3C,E). These data indicate that: (i) activation of α3β4-containing nAChRs by neonicotinoids drives a secretory response at high concentrations from adrenal medulla glands ex vivo; and (ii) a paradoxical effect develops in the presence of NIC, which may be due to the competition of NIC and neonicotinoids on nAChRs. A disruptive effect of neonicotinoids on the physiological secretion in the adrenal medulla at low doses could then be anticipated.

### 2.3. Effects of Neonicotinoids on ABP in Rat

To evaluate the acute effects of neonicotinoids in vivo, ABP was continuously measured through a transducer inserted in the femoral artery of anesthetized rats (Figure 4A). Different control experiments were performed to test the adequate response of rats in this paradigm (Appendix A). Intravenous injection (i.v.) of AngII (hypertensive) significantly increased mean ABP whereas i.v. injection of ACh (hypotensive) caused a significant decrease of mean ABP, as expected (Figure 5A,B and Appendix A). Administration of the vehicle solution, or saline, did not result in any modification. These four control injections were systematically carried out before every assay (Figure 5A,B and Appendix A). To assess the intoxication protocol, we designed a negative control experiment (CTRL group) where rats received an injection of physiological saline with increasing DMSO concentration (2% maximum): no significant change in ABP was observed (Figure 5A,B), suggesting that our protocol did not influence the vascular parameters and allowed us to challenge rats with NIC ± ACE or CLO through an acute exposition.

When rats received either CLO or ACE injections, the ABP variation of each rat remained close to 0 mmHg which did not significantly differ from the CTRL experiment (Figure 5A,B). Acute exposition to neonicotinoids alone influenced ABP differently to NIC, which exerted its hypertensive effect as expected (mean ± SEM: 51 ± 7 mmHg; range 39–69 mmHg), which is not significantly different from the AngII-induced ABP increase (Figure 5C,D). Because NIC and neonicotinoids interact on the same molecular target, i.e., nAChR, we next challenged both molecules on rat ABP to analyze a possible cocktail effect on this vascular parameter: a fixed dose of neonicotinoid (ACE 0.093 mg·kg${}^{-1}$ or CLO 0.33 mg·kg${}^{-1}$) was injected simultaneously with increasing doses of NIC (Figure 5C,D). A non-expected biphasic response of rat ABP is observed: an initial ABP decrease (ACE: mean ± SEM −17 ± 5 mmHg; range −11 to −12 mmHg; CLO: mean ± SEM −12 ± 6 mmHg; range −1 to −13 mmHg) is followed by an ABP increase (ACE: mean ± SEM 52 ± 7 mmHg; range 31 to 71 mmHg; CLO: mean ± SEM 50 ± 9 mmHg; range 24 to 85 mmHg). Thus, NIC together with ACE or CLO seemed to deeply modify the kinetics of ABP variation, exerting a non-expected hypotensive effect before the expected ABP increase. These data indicate that neonicotinoids impair the ABP when challenged with a full agonist of nAChRs.

### 2.4. Modeling of Neonicotinoid–α3β4AChR Interaction

To gain further insight into the mechanism of action of neonicotinoids, we modeled the interaction of CLO and ACE on the rat α3β4 nAChR. We first generated a model of the rat α3β4 nAChR by homology modeling (Figure 6). We then docked NIC, CLO and ACE on the NIC binding site at the extracellular domain of the receptor between α and β subunit, i.e., at the orthosteric site (Appendix A) [25]. The three molecules were found to interact with the residues known to contribute to the binding site from so-called loops A, B, C, D and E (Figure 6), i.e., αY93; αY190; αC192; αC193; αY197 and αW149 [45]. These results are therefore consistent with the hypothesis that these molecules bind to the orthosteric binding site of nAChRs [45,46].

## 3. Discussion

Initially designed to act as NIC with less mammalian toxicity, neonicotinoids are synthetic molecules sharing the same molecular target with its molecular template, i.e., the nAChR. At insect cholinergic synapses, they mimic ACh without being degraded and this interference leads to the continuous activation of nAChRs, responsible for their lethal effects [11]. For several years now considering the wide use of these compounds throughout the world and their alarming persistence in environment, studies have focused on their effects in vertebrates, especially mammals, where they exert neurotoxic effects [48,49,50,51,52] and endocrine disruption [24,53,54,55,56,57]. The mode of action of neonicotinoids in vertebrates, particularly in mammals, is imperfectly described and needs to be better understood, to fully figure out the consequences on human health and to anticipate the occurrence of chronic pathologies. Neonicotinoids can cross several biological barriers –including the digestive tract and the blood–brain barrier. Their low molecular weight and lipophilic profile allow them to access the whole body and exert acute effects for several hours according to pharmacokinetic studies. When exposed to CLO (20 mg·kg${}^{-1}$) through the intraperitoneal route, mice exhibited a peak concentration of 17, 55 and 14 ppm in brain, liver and plasma, respectively, after 15 min [58]. CLO reached an undetectable level in these 3 compartments after 240 min. After mouse oral exposure, ACE (71 and 710 mg·kg${}^{-1}$·day${}^{-1}$ for 3 or 7 days) accumulates in various amounts in the brain [59]. In a recent study, deuterium-labeled neonicotinoids (including CLO and ACE) were orally ingested by healthy adults [60]. Both molecules could be detected in urine for 4 consecutive days after exposure. The average amounts excreted were 1.14 g·d${}^{-1}$ for desmethyl-ACE and 0.51 μg·d${}^{-1}$ for CLO.

In our study, we first assessed the ability of CLO and ACE to activate α3β4 nAChRs, which are mainly expressed in the rat peripheral nervous system [61]. Both molecules acted as weak agonists of this receptor, eliciting currents in Xenopus oocytes much smaller than currents recorded in response to ACh or NIC. The low currents induced by CLO or ACE indicate a low affinity and a low efficiency compared to the natural agonists. This is comparable to previous data showing a weak agonist effect of CLO and imidacloprid (IMI) on the human α2β4 nAChR [41]. The concentrations of neonicotinoids used (3–300 μM) were close to what we tested in our study (0.12–20 μM). Similar data using IMI and ACE are available on rat cerebellum neurons, containing nAChRs expressing α3, α4 and α7 subunits. In these cells, the addition of 1–100 μM ACE or IMI evoked intracellular Ca${}^{2+}$ influx which can be inhibited by selective nAChRs antagonists [48]. Unexpectedly, CLO and ACE were recently shown to have strong agonist effects–with EC${}_{50}$ in the range of ACh–on rat α7 nAChRs [42]. In addition, these neonicotinoids are likely to act as positive allosteric modulators of this α7 nAChR. Further electrophysiological characterization is required to define on which nAChRs neonicotinoids act potently. We cannot rule out the fact that these compounds also target other nAChR subtypes since cholinergic networks are largely distributed in the whole body [62]. As of today, it would be difficult to precise the role of each nAChR subtype in the cardiovascular consequences of neonicotinoid acute exposure. Nevertheless, the role of the α3β4 receptor has been evidenced in the nicotine hypertensive effect [35] and might be a target of interest for neonicotinoids.

The ability of neonicotinoids to activate nAChRs and their potential role as endocrine disruptors prompted us to consider their effects on the peripheral nervous system. Since α3β4 receptors are widely expressed at cholinergic synapses of the adrenal medulla and responsible for the initiation of catecholamine secretion [43,44], we tested the capacity of neonicotinoids to impact the secretory response ex vivo. Our data indicate that CLO and ACE promote epinephrine secretion at much higher concentrations than NIC does. These secretory effects for both neonicotinoids are likely to occur through α3 subunit-containing nAChRs because of their strong inhibition by HEX, but we cannot rule out the fact that several nAChR subtypes might be involved in this effect. We observed no significant effect with 100 μM CLO or ACE assuming their weak agonist effect on nicotinic receptors.

Our results are consistent with previous works, where neonicotinoids, particularly CLO, have been shown to stimulate catecholamine secretion. Indeed, at millimolar concentrations, CLO induces secretory effects on dopamine release in vivo through the activation of α4β2 and α7 nAChRs subtypes in rat striatum [63,64,65]. IMI—which activates nAChRs in secretory PC12 cells [66]—produces a significant increase in serum epinephrine and norepinephrine levels in adult rats at 1 mg·kg${}^{-1}$ dose, leading to behavioral impairments [67]. Such increase is comparable to what we found here and suggest that neonicotinoids could be considered as endocrine disruptors, consistently with their pharmacokinetic properties. Such endocrine effects could be the consequence of their agonist action on nAChRs. Indeed, the activation of nicotinic receptors promote an increase of intracellular Ca${}^{2+}$ and a membrane depolarization, leading to the release of catecholamine. However, NIC seems much more efficient at stimulating catecholamine biosynthesis [68] and secretion (our present data). Interestingly, an unexpected competition developed between NIC and neonicotinoids (Figure 2), suggesting that neonicotinoids might impair the binding of NIC or ACh to the nAChR binding site [66]. Consistent with their weak agonist effect on nAChRs, neonicotinoids could compete with ACh in adrenal synapses to lower the level of ACh-induced epinephrine secretion.

The acute and chronic effects of NIC on the cardiovascular system are known, and have been the subject of numerous studies, particularly associated to tobacco smoking [69,70]. These effects are mediated by NIC action on the CNS and autonomic PNS, where it promotes the release of norepinephrine and on the adrenal medulla, in which it contributes to epinephrine secretion [71,72]. With a broad presence of nAChRs in the mammalian nervous system, neonicotinoids may target cholinergic networks and cause deleterious effects. Acute poisoning after ingesting or inhaling a large quantity of neonicotinoids induces various symptoms in humans: respiratory decompensations, disorders of consciousness, muscle disorders and impairment of cardiovascular function with tachycardia and bradycardia and arterial hypotension [31,73,74]. A 74-year-old woman who ingested 100 mL of insecticides containing 2% ACE showed muscle weakness, hypothermia followed by cardiac arrhythmia and hypotension [75]. The hypertensive effect of NIC that we observed in anesthetized rats is consistent with previous works [76,77]. An acute NIC injection in rat induces elevated ABP which can be attenuated by mecamylamine, a nonselective nicotinic receptor antagonist and selectively blocked by hexamethonium, the selective α3-containing nicotinic receptor antagonist [35]. The main effect of NIC on ABP then occurs through the α3β4 nicotinic receptor. ACE and CLO did not modify ABP when injected alone, but it produced hypotension in co-application with NIC, followed by a large increase in ABP. The lack of effect of neonicotinoids alone can be correlated with the pharmacological properties of ACE/CLO, as weak agonists of nAChRs, but we cannot rule out that our protocol of anaesthesia could have affected the effect of neonicotinoids, because the combination of ketamine/xylazine is known to exert neurotoxic effects [78]. Cytisine, also a partial nAChR agonist used as a smoking cessation treatment, stimulates nAChRs in the PNS and the adrenal gland. As such, cytisine promotes catecholamine secretion, with cardiovascular consequences [79]. Its action is comparable to NIC, although more moderate. Alternatively, varenicline, also used for smoking cessation, is a partial nAChR agonist that competitively inhibits NIC binding to these receptors [80]. Chronic oral treatment of Wistar rats with varenicline induced a decrease of ABP [81]. These data clearly indicate that nAChR agonists directly or indirectly disrupt vascular function and produce ABP variations. As such, neonicotinoids are quite likely to produce arterial dysfunctions, through their action on nAChRs in the autonomous nervous system. The biphasic response we observed with neonicotinoids added to NIC consists of a decrease (hypotensive effect) followed by an increase (hypertensive effect) of ABP. Acute arterial hypotension is a consequence of: (i) a blood pressure decrease; and/or (ii) a decrease of secretion/release of catecholamine [47]. This BP decrease can be the consequence of a muscarinic effect, through the stimulation of nAChRs of the parasympathic system. Such a biphasic response (hypotension followed by hypertension) has been observed with several neurotoxins. Brevetoxin, for instance, is a Na${}^{+}$ channel activator acting on the autonomous nervous system and the adrenal medulla, causing a decrease followed by an increase of ABP in mammals [82].

Our modeling of ACE/CLO interaction with the α3β4 receptor is coherent with the notion that neonicotinoids bind to the orthosteric site of nAChR. Most amino acids implicated in the binding site of NIC are also involved in the binding site of neonicotinoids. It is possible that there are slight differences, but care must be taken in our interpretation; as we performed a homology modeling and then a docking, each of them may introduce uncertainties. In our model the residues known to be part of the binding site are shown [83]. These are also the same residues which have been found on structures of NIC bound to the α4β2 receptor (PDB entry 6CNK, 6CNJ, 5KXI) and to the α3β4 receptor (PDB entry 6PV7). In these structures, we find the cation/π bond with the trp of loop B, as well as the proximity of the residues of loops C, A, D and E. This is also coherent with recent studies of the interaction of neonicotinoids with the ACh/NIC binding site on AChBP [84,85]. We conclude that the binding site is identical for NIC and for neonicotinoids, the cation/π interaction being a strong marker of NIC (and ACh) binding to nAChRs. However, the fact that neonicotinoids and NIC bind to the same site with different agonist effects is not exceptional, since it has been observed with other agonists. For instance, lobeline–which is considered a low agonist and an antagonist of nAChRs–and ACh bind to the orthosteric site of AChBP from *Aplysia californica* [86]. In our case, it would be speculative to propose that the difference lies more in the nature of the ligand than in the residues with which it interacts. It could be hypothesized that neonicotinoids could bind slightly higher, but we do not have any evidence to support it.

## 4. Materials and Methods

### 4.1. Animal Models and Experimental Conditions

Five-month-old Wistar Kyoto male rats (Envigo, Gannat, France) weighing 350–450 g were acclimatized in a temperature and light controlled-room (23 ${}^{\circ }$C; 12L:12D), in translucent cages (2–5 rats per cage, dimensions 43 cm × 27 cm × 16 cm), with food and water ad libitum. All experiments conducted randomly were carried out by a single female experimenter, to avoid stress. Four to six animals were used in each experimental group. Adult female *Xenopus laevis* were purchased from CRB (Rennes, France) and were bred in the laboratory animal facility in accordance with the recommendations of the EU Directive 2010/63/EU on the care and use of laboratory animals. All anesthetized animals recovered after 2–3 h. The protocols involving living animals have been approved by the regional ethics committee (https://www.ceea-paysdelaloire.com, visited on 11 May 2021 APAFIS file 7586 for rats and N° CEEA.2012.68 for Xenopus) and by the French Ministry of Agriculture.

### 4.2. Solutions and Drugs

Acetylcholine chloride (ACh), nicotine tartrate salt (NIC), HEX, CLO and ACE were purchased from Sigma-Aldrich (St-Quentin Fallavier, France). Angiotensin II (AngII), ketamine (Imalgene^®^ 1000) and xylazine (Rompun^®^ 2%) were purchased from Bachem (Bubendorf, Switzerland), Merial (France) and Bayer (France), respectively.

### 4.3. Molecular Cloning of Rat nAChR Subunits

Total RNA were extracted from Wistar male rat adrenal glands and converted into cDNA by reverse transcription according to the manufacturers’ instructions (RNeasy micro kit: Qiagen, Courtaboeuf, France; Superscript^®^III First-strand Synthesis Super mix, Invitrogen, Villebon-sur-Yvette, France). The complete open reading frame of rat nAChR α3 and β4 subunits were amplified using gene-specific primers designed from reference sequences (Genbank accession number NM_052805 (α3), NM_052806 (β4), Appendix A). Nested PCR amplifications were conducted as follows: 95 ${}^{\circ }$C for 5 min; 30 cycles of 95 ${}^{\circ }$C for 30 s and 68 ${}^{\circ }$C for 1 min 30; and 68 ${}^{\circ }$C for 4 min. PCR fragments were purified according to the manufacturer’s instructions (Nucleospin® Gel and PCR clean-up, Macherey-Nagel, Germany). The full-length cDNAs were flanked with EcoRI and HindIII restriction sites to ensure the directional cloning into the vector pGEM-HE (kindly provided by Professor Pongs, Institute for Neural Signal Transduction, Hamburg, Germany). Recombinant plasmids were screened before sequencing (GATC Biotech, Constance, Germany). Sequence analysis was performed using blast algorithm on NCBI database (https://blast.ncbi.nlm.nih.gov/Blast.cgi visited on 11 May 2021; [87]; Appendix A). cRNA of each subunit were synthesized in vitro from linearized recombinant plasmid using mMESSAGE mMACHINE^®^ transcription kit according to the manufacturer’s instructions (Ambion) and cRNA concentration was assessed using spectrophotometer (Nanodrop™ 2000/2000C Thermofisher Scientific Inc., Waltham, MA, USA).

### 4.4. Expression of α3β4AChRs in Xenopus Oocytes

Oocytes were harvested and injected as previously described [88]. Briefly, ovarian lobes were taken from female Xenopus (*Xenopus laevis*) anesthetized in Tricaine (0.15% in ice-cold water) for 20 min. Oocytes were harvested and placed in standard oocyte saline (SOS containing 100 mM NaCl, 2 mM KCl, 1.8 mM CaCl${}_{2}$, 1 mM MgCl${}_{2}$, 5 mM HEPES, pH 7.4). Stage 5–6 oocytes were partially defolliculated by enzymatic treatment with 2 mg·mL${}^{-1}$ collagenase (type IA, Sigma) in Ca${}^{2+}$-free SOS for 60 min and eventually manually defolliculated in SOS medium. To express functional nAChRs, a volume of 13.8 nL of α3β4 nicotinic receptor mRNA (1:1—at a concentration of 988 ng·L${}^{-1}$) was injected into the cytoplasm of individual defolliculated oocytes, using an automatic nanoinjector (Nanoject II Drummond Scientific). The stoichiometric arrangement of this receptor has been shown to exert minimal consequences on its pharmacological properties [89]. Subsequently, oocytes were incubated in the incubation medium (autoclaved normal SOS supplemented with gentamycin (50 μg·mL${}^{-1}$), penicillin (100 UI·mL${}^{-1}$), streptomycin (100 μg·mL${}^{-1}$) and sodium pyruvate (2.5 mM) for 24 h at 18 ${}^{\circ }$C.

### 4.5. Two-Electrode Voltage Clamp Recording

The technique was previously described [88,90]. The expression of nAChR in oocytes was tested at a holding potential of ™60 mV using a TEV-200 amplifier (Dagan Corporation, Minneapolis, USA). Digidata 1440A interface (Axon CNS Molecular Devices, CA, USA) and pCLAMP™ 10 software (Molecular Devices) were used for current recording. Injected oocytes were continuously superfused with SOS medium at room temperature and were challenged with drugs in SOS. Stock solutions were prepared as follows: ACh was dissolved in distilled water at 100 mM, while NIC and ACE/CLO were dissolved in DMSO at 1 and 0.1 M (solubility limit), respectively. All dilutions were made in normal SOS. Electrodes were filled with 1 M KCl/2 M K-acetate and display typical resistances of 0.5–2 MΩ in SOS. Control experiments were performed using DMSO 1%. To evidence the expression of α3β4 nAChRs in oocytes, we challenged injected and non-injected oocytes with ACh, NIC, ACE or CLO (Appendix A). Non-injected oocytes did not develop any current. Data were analyzed using pCLAMP 10 software. Mean current amplitudes were calculated from at least five different cells from two different oocyte batches and processed as current density (nA/nF). Concentration–effect relationships were analyzed using the following equation:$$Y=Y_{min}+\left(Y_{max}-Y_{min}\right)/\left(1+10^{\left((LogEC50-X)*nH\right)}\right)$$
where *X* is the concentration of agonists, *Y*${}_{min}$ and *Y*${}_{max}$ are the minimum and highest responses, EC${}_{50}$ is the half maximal effective concentration and nH is the Hill coefficient.

### 4.6. Epinephrine Secretion Assay

Epinephrine secretion was monitored from acute adrenal slices, as previously described [44]. Briefly, acute slices (150 μm thickness) from left and right adrenal glands in Wistar male rats were cut using a vibratome (DTK-1000, D.S.K, Dosaka EM CO. LTD, Kyoto, Japan). The slices recovered 15 min in Ringer’s saline (125 mM NaCl, 2.5 mM KCl, 2 mM CaCl${}_{2}$, 1 mM MgCl${}_{2}$, 1.25 mM NaH${}_{2}$PO${}_{4}$, 26 mM NaHCO${}_{3}$, 12 mM glucose and buffered to pH 7.4) in a perfusion chamber (37 ${}^{\circ }$C, 95% O${}_{2}$/5% CO${}_{2}$) before being challenged with NIC/neonicotinoid (ACE or CLO)/hexamethonium (HEX), an α3-containing nAChR antagonist. The basal epinephrine secretion (B) was measured in the supernatant of a 5 min-bath of each adrenal slice containing either Ringer’s saline or NIC (10 μM). Each slice was transferred in a test tube and then challenged for 5 min with a solution containing NIC (1, 3, 10 and 100 μM), ACE (100 and 500 μM) or CLO (100 and 500 μM), alone or in combination (NIC+CLO or NIC+ACE), with or without HEX (200 μM), before removing the supernatant for epinephrine measurement referred as test epinephrine secretion (T). HPLC-based epinephrine assay was conducted as previously described [44]. The results are expressed as T/B ratio of epinephrine release. The concentrations of neonicotinoids used in this ex vivo assay—100 and 500 μM—can be approximately equivalent to 30 and 150 mg·kg${}^{-1}$ in vivo, respectively.

### 4.7. Intoxication Protocol and ABP Measurement

Wistar male rats were anesthetized by intraperitoneal injection of ketamine at a dose of 80 mg·kg${}^{-1}$ and xylazine as analgesic at 15 mg·kg${}^{-1}$. We are aware that this anesthetic combination could interfere with our data [91,92] since ketamine/xylazine are known to disrupt cardiovascular parameters. To minimize the anesthetic side effects, we monitored ABP throughout the experiment and verified, before each injection, that the mean ABP is within the range of physiological values for Wistar-Kyoto rats (94.18 ± 2.14 mmHg, Appendix A). For the measurement of ABP via arterial catheterization, the anesthetized animal was positioned back on a heating mat (maintained at 38 ${}^{\circ }$C). The PE10 catheter (DI: 0.28 mm, OD: 0.61 mm) welded to PE50 (DI: 0.58 mm, DE: 0.965 mm) is placed at the level of the right femoral artery and is connected to the pressure sensor. The measurements are collected using a Biopac data acquisition system (Biopac systems MP100 coupled to AcqKnowledge^®^ software). Injections are made by the second PE50 catheter placed at the level of the left femoral vein (Figure 4A). Based on the knowledge of neonicotinoid molecules, we estimated that an intravenous dose of 1 mg·kg${}^{-1}$ CLO or ACE is equivalent to a concentration of 5–20 μM, for which we found a weak agonist effect of these molecules in electrophysiology. The following injections were performed subsequently in each animal. Firstly, AngII (10 ng·kg${}^{-1}$), ACh (0.5 μg·kg${}^{-1}$), vehicle solution (DMSO 2% and NaCl 0.9%) and physiological saline (NaCl 0.9%) were administered as positive and negative control molecules for each animal. Depending of their group assignment (NIC, ACE, CLO and CTRL), a selected dose of NIC, ACE, CLO or DMSO (2%) was injected into the femoral vein of each rat with a 4–5-min interval between two injections (NIC: 3 × 10${}^{-1}$ mg·kg${}^{-1}$; ACE/CLO: 1 mg·kg${}^{-1}$) (Figure 4B). To study the effect of both molecules, NIC was administered to two other rat groups (NIC+ACE and NIC+CLO), associated with a single dose of either ACE (0.1 mg·kg${}^{-1}$) or CLO (0.33 mg·kg${}^{-1}$). These fixed doses of neonicotinoids were adjusted from the Acute Reference Dose (ARfD) promoted by FAO/WHO (reviewed in [93]). The dose for CLO was limited to 0.33 mg·kg${}^{-1}$ (and not 0.6 mg·kg${}^{-1}$) because we could not exceed the threshold limit for DMSO in vivo. A negative control was also performed with vehicle solution injections on a time scale similar to that of intoxications (Figure 4B). Femoral ABP was recorded continuously (1000 Hz) for 1 h and the animals were killed by exsanguination. Mean ABP was measured on recordings, over a 2-s period before and after the injections using AcqKnowledge^®^ software (Biopac, Goleta, CA, USA). The difference between these mean values (ΔABP) was used for statistical analysis (Figure 4C).

### 4.8. nAChR Homology Modeling and Ligand Docking

The chosen template was that of the human α3β4 nAChR obtained in complex with NIC (PDB code 3PV7). The sequence of the template was aligned to that of rat α3 and β4 with the software T-Coffee [94]. The alignment was edited manually to remove the long stretch of α3 and β4 residues for which there is no counterpart in the template, i.e., N-terminus, C-terminus and M3-M4 loop. The resulting alignment was used, together with the PDB structure 3PV7, to produce the target model with Modeler version 9.19 [95]. The automodel method was used with “very slow” optimization level and three repetitions. In total, 100 models were prepared and the best, according to the DOPE energy function, was selected. The structures of ligands were retrieved from Pubchem as sdf files, i.e., Clothianidin (86287519) and Acetamiprid (213021). The structure of the protein and ligand were converted to pdbqt files with the software Open Babel 2.4.1 [96]. Docking was performed with the software smina [97]. The docking box was defined automatically using the nicotine found in the experimental structure.

### 4.9. Data Analysis

Statistical analysis were performed using GraphPad Prism^®^ 7 software (La Jolla, CA, USA). Electrophysiological data were analyzed using a nonlinear regression for each treatment concentration–effect relationship. The difference in current amplitude among treatment was assessed using one way ANOVA and Tukey’s multiple comparison post hoc test. Epinephrine secretion data (i.e., NIC/ACE/CLO dose–effect, α3β4 role in NIC or ACE-CLO-induced epinephrine secretion, ACE/CLO effect on NIC-induced epinephrine secretion) were assessed by Kruskal–Wallis one-way ANOVA followed by Dunn’s multiple comparison post hoc test. ABP data (i.e., CLO/ACE/NIC effect on ABP compared to hypertensive (AngII), hypotensive (ACh) or vehicle solution (VEH) as well as to the CTRL solution, ACE/CLO effect on NIC-induced hypertensive effect) were assessed by two-way ANOVA and Tukey’s multiple comparison post hoc test. Data are mean ± SEM. In Figure 2, Figure 3 and Figure 5, the different lowercase letters indicate significant differences between treatments. Any two means that do not share the same letter are significantly different (*p* < 0.05) after running the ANOVA analysis. The lowercase letters a–d are used to indicate which means differ and which ones do not based on the multiple comparison post hoc tests.

## 5. Conclusions

Since their first introduction in the early 1990s, neonicotinoid insecticides have been used extensively to control harmful insects and increase agricultural productivity. The preoccupation for the human health has emerged recently [98]. Our data show that the neonicotinoids CLO and ACE, through a weak stimulation of nAChRs, elicit epinephrine secretion with no apparent disturbance of rat ABP in vivo. However, ACE or CLO modified the kinetics of NIC-induced hypertensive effect. This raises concern about sub-chronic and chronic effects of these insecticides on the human health.

## Figures and Tables

**Figure 1 ijms-22-05106-f001:**
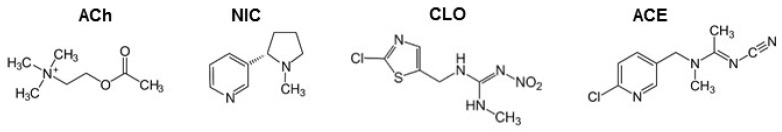
Molecular structures of acetylcholine (ACh), nicotine (NIC) and the neonicotinoids clothianidin (CLO) and acetamiprid (ACE). NIC, the molecular template of most neonicotinoids, is protonated at physiological pH, whereas CLO and ACE possess an electronegative nitro- and cyano- functional group, respectively [25].

**Figure 2 ijms-22-05106-f002:**
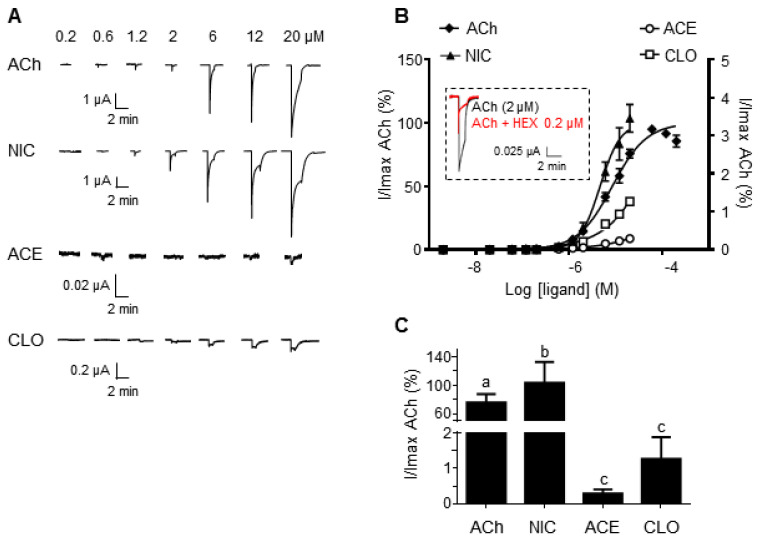
Pharmacological profile of ACh, NIC, ACE and CLO on rat α3β4 nAChRs expressed in Xenopus oocytes. (**A**) ACh (*n* = 12) and NIC (*n* = 7) elicit robust currents, while ACE (*n* = 8) and CLO (*n* = 7) induce much more modest currents. Due to the solubility limit in DMSO, ACE and CLO could not be used at concentrations upper than 2 × 10${}^{-5}$ M. (**B**) Concentration–response curves of ligand-evoked currents which are expressed as a % of ACh–elicited current amplitude. Two Y axis were used to visualize ACh and NIC (left), and CLO and ACE (right). Inset: Hexamethonium (HEX, 0.2 μM) an α3-containing nAChRs antagonist, inhibits ACh-elicited current as expected. (**C**) Peak currents (±SEM) elicited by 20 μM ACh, NIC, ACE or CLO on rat α3β4 nAChRs. Different lowercase letters above the graphs indicate significant differences between treatments according to Tukey’s multiple comparison post hoc test (*p* < 0.05).

**Figure 3 ijms-22-05106-f003:**
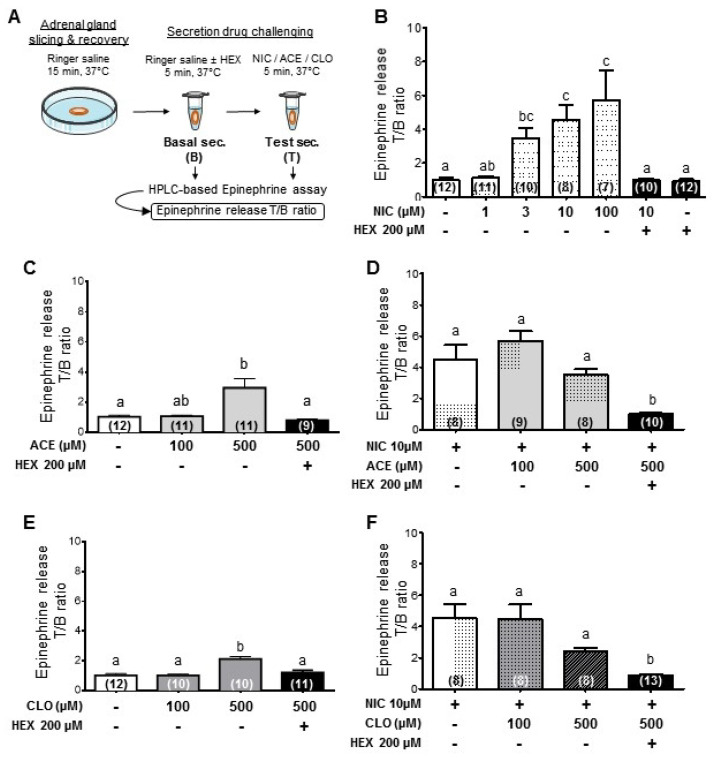
Influence of NIC, ACE and CLO added separately or in combination on epinephrine release from rat adrenal gland slices. (**A**) Experimental design modified from [44]. (**B**) Concentration–response of epinephrine release following NIC stimulation ± HEX, an α3-containing nAChR antagonist. (**C**–**F**) Epinephrine release following either ACE or CLO treatment in combination or not with NIC ± HEX. Data are mean ± SEM and values in brackets correspond to the number of biological replicates. Different lowercase letters above the graphs indicate significant differences between two histograms within each graph according to Dunn’s multiple comparison post hoc test (*p* < 0.05). Note that neonicotinoids are able to stimulate the secretion of epinephrine when applied to the medulla tissue alone.

**Figure 4 ijms-22-05106-f004:**
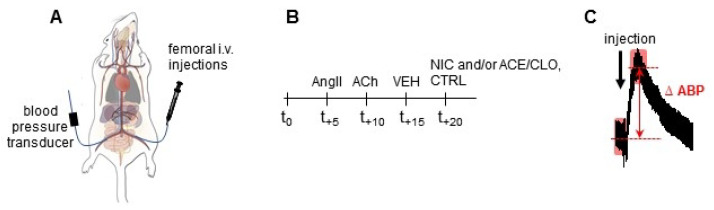
Schematic protocol of rat arterial blood pressure (ABP) measurements. (**A**) Illustration of an anesthetized rat with a catheter inserted in right femoral artery for ABP measurements through a blood pressure transducer, and with a catheter inserted in left femoral vein for i.v. injections. Rat drawing modified from Watts et al. [47]. (**B**) Schematic time-course of the protocol. Each rat was challenged every 5 min with AngII, ACh and vehicle solution (VEH) before fixed concentrations of NIC, ACE, CLO or DMSO 2% (CTRL). (**C**) Analysis method of raw trace to collect the difference between mean arterial blood pressure (ΔABP). See Materials and Methods Section 4.7 for further explanation.

**Figure 5 ijms-22-05106-f005:**
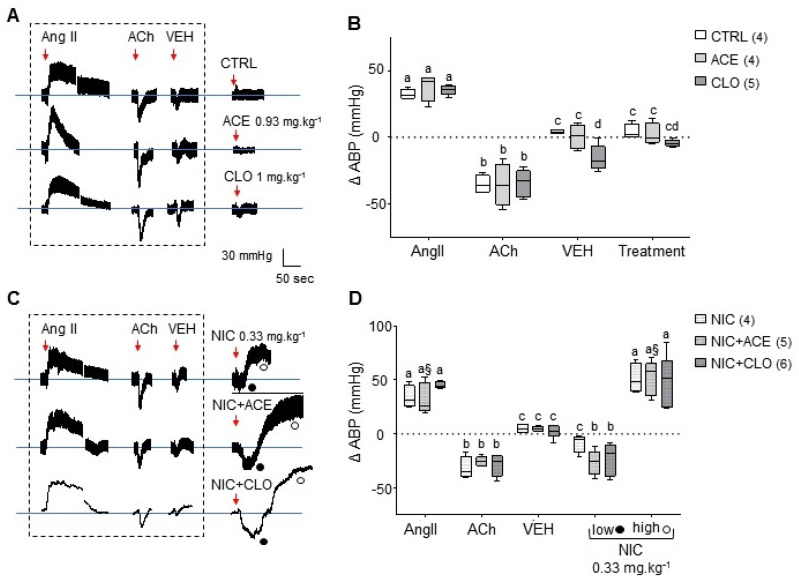
Influence of NIC, ACE and CLO on rat ABP. (**A**) Representative raw traces of ABP following ACE or CLO infusion (red arrow with doses used) compared to the CTRL group. As expected, protocol assessment molecules such as AngII (1.05 × 10${}^{-5}$ mg·kg${}^{-1}$) and ACh (0.5 × 10${}^{-5}$ mg·kg${}^{-1}$) induce increased and decreased ABP, respectively. Neither DMSO 0.5–3% (VEH) nor ACE or CLO affected ABP, whatever the dose tested. (**B**) Box and whisker plots of ΔABP data in the CTRL, CLO and ACE groups following i.v. injections, 0.93 mg·kg${}^{-1}$ for ACE and 1 mg·kg${}^{-1}$ for CLO. (**C**) Representative raw traces of ABP following infusions of NIC alone or NIC with either ACE or CLO. NIC was used at 0.33 mg·kg${}^{-1}$ (red arrow) in combination with a single dose of ACE (0.09 mg·kg${}^{-1}$) or CLO (0.33 mg·kg${}^{-1}$). AngII, ACh and VEH infusions triggered BP response as expected. In addition, 0.33 mg·kg${}^{-1}$ NIC induced a BP increase as expected, while its combination with either ACE or CLO was responsible for a biphasic response with low (o) and high (•) ABP. (**D**) Box and whisker plots of ΔABP data in the NIC, NIC+ACE and NIC+CLO groups following the various infusions cited above. Whiskers are minimum to maximum values and bar represents the median value. Different lowercase letters and § above the graphs indicate significant differences between treatments according to Tukey’s multiple comparison post hoc test (*p* < 0.05).

**Figure 6 ijms-22-05106-f006:**
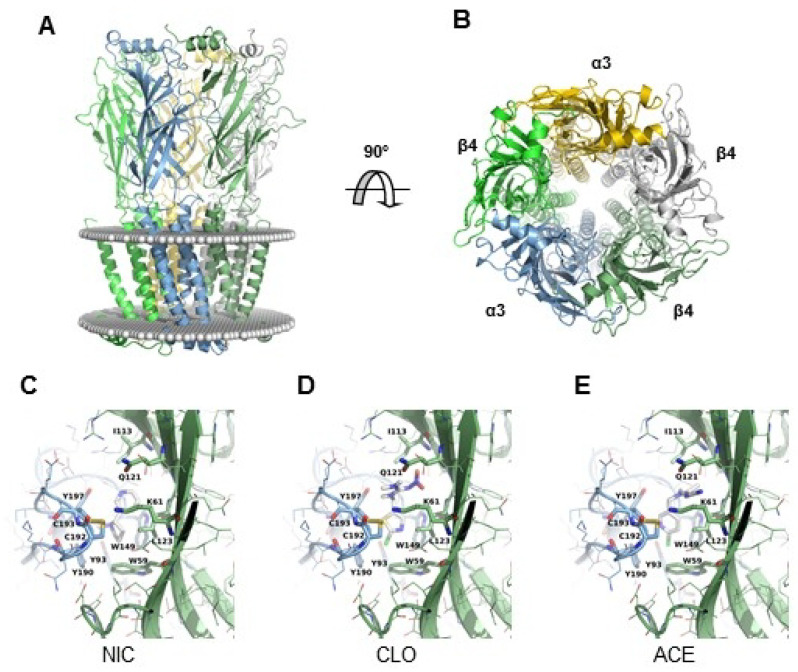
Homology modeling of rat α3β4 nAChR and molecular docking of NIC, CLO and ACE. (**A**,**B**) Model of the receptor viewed from the membrane plane (**A**) and from the extracellular domain (**B**). The protein is shown in cartoon representation with a different color code for each polypeptide. The position of the membrane is represented by spheres. (**C**–**E**) Graphical representations of the orthosteric site of rat α3β4 nAChR. The α3β4 subunit interface in complex with: NIC (**C**); CLO (**D**); or ACE (**E**).

## Data Availability

All the data are contained within the article.

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
