# Peer review of "Partial Agonist Activity of Neonicotinoids on Rat Nicotinic Receptors: Consequences over Epinephrine Secretion and In Vivo Blood Pressure"

_ijms, 2021, doi:10.3390/ijms22105106_

Round 1
Reviewer 1 Report
The work by Park J et al. reports a detailed and well-executed analysis of potential harmful and neurototoxic effects of Neonicotinoid insecticides on the mammalian cholinergic networks, focusing on actions at the peripheral nervous system. The authors make use of several different approaches and provide compelling evidence for an effect of these compunds. However, some aspects need to be better explained or deepened in order that the manuscript can be accepted for publication. My recommendation is that the following points are addressed:
1) abstract, line 19-20. The sentence is a little bit overstated and should be adjusted to avoid claims of priority. Also, it looks to me as a transversal experimental approach, rather than a longitudinal study
2) line 74: a proof of expression of alfa3 and alfa 4 nAChR subunits in oocytes should be provided, e.g. by performing a western blot analysis. alternatively, the authors should perform the same analysis reported in Fig. 2 in non-transfected oocytes.
3) Fig. 2C statistical analysis missing
4) Fig. 3 and 5 the statistical analysis is barely visible at the resolution provided, and most importantly is not properly defined either in the relative captions and in the methods section. The authors should use a more typical asterisk-based code to mark and comment on the differences they observe.
5) Fig. 4B the timeline is nice, but the time points should be explicitly indicated to allow repeatability of the experiment.
6) Fig. 6, in panel B "extracellular domain" is more straightforward than synaptic cleft. Furthermore, I have a major concern regards the docking study reported in panel c-d-e. Is it the binding site exactly conserved among NIC, CLO and ACE (too low figure resolution to appreciate)? how do you then explain the lower affinity displayed by CLO and ACE when compared to NIC as evidenced from data reported in Fig. 2? Provided that the partial agonism demonstrated in Fig. 2 can be justified in some way by the docking study, these would well fit with the epineprhine secretion experiment. But how do you then explain the unexpected biphasic response seen in the ABP assay? These data suggest a more complex interplay, than a simple competition to the same binding site. This would indeed result in a simple lower increase of BP, exaclty as detectd for the epineprhine secretion. Some temptative explanations should be provided in the discussion section. Lines 271-275 should be reviewed in light of these observations.
7) line 202: which receptor do the authors refer to?
Author Response
We wish to thank Reviewer 1 for its helpful and accurate comments. We answered all the questions addressed and modified the draft accordingly.
Reviewer 1
The work by Park J et al. reports a detailed and well-executed analysis of potential harmful and neurotoxic effects of Neonicotinoid insecticides on the mammalian cholinergic networks, focusing on actions at the peripheral nervous system. The authors make use of several different approaches and provide compelling evidence for an effect of these compounds. However, some aspects need to be better explained or deepened in order that the manuscript can be accepted for publication. My recommendation is that the following points are addressed:
1) abstract, line 19-20. The sentence is a little bit overstated and should be adjusted to avoid claims of priority. Also, it looks to me as a transversal experimental approach, rather than a longitudinal study
Answer: We modified the abstract as requested by Reviewer 1.
2) line 74: a proof of expression of alfa3 and alfa 4 nAChR subunits in oocytes should be provided, e.g. by performing a western blot analysis. alternatively, the authors should perform the same analysis reported in Fig. 2 in non-transfected oocytes.
Answer: Instead of using low specificity antibodies (Santa Cruz sc-5591), control experiments were performed using non-injected oocytes. The expression of α3β4 nAChRs in oocytes was evidenced using injected and non-injected oocytes challenged with ACh or NIC. Non-injected oocytes did not develop any current. We added current traces in a supplementary figure S1.
3) Fig. 2C statistical analysis missing
Answer: We performed one-way ANOVA and Tukey’s multiple comparison for this figure. Statistical differences were properly added on the bars and were explained in the figure’s caption.
4) Fig. 3 and 5 the statistical analysis is barely visible at the resolution provided, and most importantly is not properly defined either in the relative captions and in the methods section. The authors should use a more typical asterisk-based code to mark and comment on the differences they observe.
Answer: The size of the statistical signs was increased to be readable. We have chosen the lowercase-based code for statistical differences to illustrate all comparisons between conditions within each experiment. It appears in the captions’ figures and in the Data analysis section (Material and Methods). Using a typical asterisk-based code does not allow to show all pairwise comparison.
5) Fig. 4B the timeline is nice, but the time points should be explicitly indicated to allow repeatability of the experiment.
Answer: The timeline was corrected according to the reviewer’s request.
6) Fig. 6, in panel B "extracellular domain" is more straightforward than synaptic cleft.
Answer: we modified the caption of Fig. 6 accordingly.
Furthermore, I have a major concern regards the docking study reported in panel c-d-e. Is it the binding site exactly conserved among NIC, CLO and ACE (too low figure resolution to appreciate)?
Answer: The bibliography on the interaction of neonicotinoids and nAChRs clearly show that these insect-selective molecules act with high affinity on insect nAChRs – explaining their high toxicity on insects – whereas they exhibit low affinity to mammalian nAChRs and low toxicity. This interaction has been described at the molecular level (Tomizawa et al., 2000 - doi: 10.1021/jf000873c.). On the other hand, NIC possesses a high affinity towards mammalian nAChRs, and low affinity for insect receptors. To date, no other molecular target has been proposed for neonicotinoids.
Our modelling of ACE/CLO interaction with the α3β4 receptor is coherent with the notion that neonicotinoids bind to the orthosteric site of nAChR, i.e the same site as ACh and NIC. Most amino acids implicated in the binding site of NIC are also involved in the binding site of neonicotinoids. It is possible that there are slight differences, but care must be taken in our interpretation as we performed a homology modelling and then a docking, each of them may introduce uncertainties.
In our model are shown the residues known to be part of the binding site (Taly et al., 2009 - doi: 10.1038/nrd2927.). These are also the same residues which have been found on structures of NIC bound to the α4β2 receptor (PDB entry 6CNK, 6CNJ, 5KXI) and to the α3β4 receptor (PDB entry 6PV7). In these structures we find the cation/π bond with the trp of loop B, but also the proximity of the residues of loops C, A, D and E. This is also coherent with recent studies of the interaction of neonicotinoids with the ACh/NIC binding site on AChBP (Alamiddine et al 2015 doi: 10.1007/s10822-015-9884-x ; Alamiddine et al 2019 doi: 10.1021/acs.jcim.9b00272). We conclude that the binding site is identical for NIC and for neonicotinoids, the cation/π interaction being a strong marker of NIC (and ACh) binding to nAChRs.
We have enlarged Figure 6 to make it more readable. We now provide a supplementary Figure Sx with a resolution sufficient to identify all residues implicated in the binding site of NIC, CLO and ACE, which are framed in red.
How do you then explain the lower affinity displayed by CLO and ACE when compared to NIC as evidenced from data reported in Fig. 2?
Answer: As we mentioned previously, NIC and neonicotinoids bind to the same site on insect and mammalian nAChRs, but with different affinities and pharmacological effects.
The fact that neonicotinoids and NIC bind to the same site with different affinities is not exceptional, since it has been observed with other agonists. For instance, lobeline – which is considered a low agonist and an antagonist of nAChRs – and ACh bind to the orthosteric site of AChBP from Aplysia californica (Taly et al. 2011). On the other hand, the binding site can change its conformation from one ligand to another (Konstantakaki et al., 2007. doi: 10.1016/j.bbrc.2007.05.126.). In our case, it would be speculative to propose that the difference lies more in the nature of the ligand than in the residues with which it interacts. It could be hypothesized that neonicotinoids could bind slightly higher, but we do not have any evidence to support it.
Provided that the partial agonism demonstrated in Fig. 2 can be justified in some way by the docking study, these would well fit with the epineprhine secretion experiment. But how do you then explain the unexpected biphasic response seen in the ABP assay?
These data suggest a more complex interplay, than a simple competition to the same binding site. This would indeed result in a simple lower increase of BP, exaclty as detectd for the epineprhine secretion. Some temptative explanations should be provided in the discussion section. Lines 271-275 should be reviewed in light of these observations.
Answer: As Reviewer 1 mentions, the low agonist effect of CLO or ACE on nAChRs can explain their low secretion induced from adrenal medulla, and their inhibitory effect on NIC-induced secretion. This is because α3β4 receptor activation drives the secretion of catecholamine by adrenal medulla (reference).
But when it comes to the arterial blood pressure, a lot more actors take place in its control. There are several ganglionic nAChRs which activation control the level of the BP, in the autonomous nervous system. We can anticipate that neonicotinoids bind to all or part of them, modulating their activity during NIC exposure.
The biphasic response we observed with both neonicotinoids added to NIC consists of a decrease (hypotensive effect) followed by an increase (hypertensive effect) of ABP. Acute arterial hypotension is a consequence of (i) a blood pressure decrease and/or a decrease of catecholamine secretion by adrenal medulla. This BP decrease can be the consequence of a muscarinic effect, through the stimulation of nAChRs of the parasympathic system. On the other hand, the decrease of NIC-induced epinephrine secretion with neonicotinoids was clearly observed in our data. This can be summarized as follows: NIC alone induces a high secretagogue effect, which is partially abolished with neonicotinoids: it ends up with a hypotensive effect.
The following hypertensive effect of the biphasic response could be the consequence of either (i) the baroreflex in response to the arterial hypotension, or (ii) the competition between neonicotinoids and NIC at the level of cholinergic synapses which control the ABP.
We now provide a new paragraph related to this question in the Discussion.
7) line 202: which receptor do the authors refer to?
Answer: α7.
Reviewer 2 Report
The manuscript of Joohee Parket al. nicely describes first longitudinal study of mammal intoxication with neonicotinoids derivatives at different scales, including in vitro, ex vivo, in vivo and in silico. Indeed, CLO and ACE, through a weak stimulation of nAChRs, elicit epinephrine secretion with no apparent disturbance of rat ABP in vivo. However, ACE or CLO
modified the kinetics of NIC-induced hypertensive effect in rats. The methods and results are described in a very clear fashion, and I very much liked the abundant clearly presented data. However, I would recommend a more detailed revision of all figures which seem to be very small for a normal publication format (two lines per page). Moreover, the writing inserted in the figures is unreadable! Check Figure 2 A, Fig. 3A, Fig. 4 & 5.
Line 123 – in vivo should be italic !
In the Materials & Methods all salts used for making saline solution should be chemically correctly CaCl2, MgCl2 (2 – as subscript) !
Congratulation to all authors !
Author Response
We wish to thank Reviewer 2 for his helpful and accurate comments. We answered all the questions addressed (in yellow).
The manuscript of Joohee Parket al. nicely describes first longitudinal study of mammal intoxication with neonicotinoids derivatives at different scales, including in vitro, ex vivo, in vivo and in silico. Indeed, CLO and ACE, through a weak stimulation of nAChRs, elicit epinephrine secretion with no apparent disturbance of rat ABP in vivo. However, ACE or CLO modified the kinetics of NIC-induced hypertensive effect in rats.
The methods and results are described in a very clear fashion, and I very much liked the abundant clearly presented data.
However, I would recommend a more detailed revision of all figures which seem to be very small for a normal publication format (two lines per page). Moreover, the writing inserted in the figures is unreadable!
Check Figure 2 A, Fig. 3A, Fig. 4 & 5.
Answer: we modified accordingly all figures mentioned by Reviewer 2 to increase their clarity and readability.
Line 123 – in vivo should be italic !
In the Materials & Methods all salts used for making saline solution should be chemically correctly CaCl2, MgCl2 (2 – as subscript) !
Answer: done.
Congratulation to all authors !
Thank you very much.
Round 2
Reviewer 1 Report
The authors made a good revision work and provided answers to most of my concerns. I still have 2 issues:
1) I recommend that the statistical analysis and especially the multiple comparisons are properly defined at least in the methods section, at paragraph 4.9. This means that the authors should define what the letter "a", "b" and "c" means, i.e. the degree of significance (p<0.05, <0.01, <0.001 etc) and to what data couples this p value refers to.
2) in the supplementary data file, I did not find the figure Sx enlarging the details of the molecular docking study reported in figure 6. I also did not find any legend of the supplementary tables or figures.
Author Response
1) I recommend that the statistical analysis and especially the multiple comparisons are properly defined at least in the methods section, at paragraph 4.9. This means that the authors should define what the letter "a", "b" and "c" means, i.e. the degree of significance (p<0.05, <0.01, <0.001 etc) and to what data couples this p value refers to.
Answer: we added the following sentencein the mat & meth section at section 4.9 Data analysis:
"Any two means that do not share the same letter are significantly different (p<0.05) after running the ANOVA analysis. The lowercase letters a, b, c and d are used to indicate which means differ and which ones do not based on the multiple comparison post-hoc tests."
2) in the supplementary data file, I did not find the figure Sx enlarging the details of the molecular docking study reported in figure 6. I also did not find any legend of the supplementary tables or figures.
Answer: We updated the Supplementary data file with the FigS2. the caption of the Figure is available at the end of the article.
We thank Reviewer 2 for all comments on the draft.
